# Temporal Trend of ST131 Clone among Urinary *Escherichia coli* Isolates in the Community: A Taiwan National Surveillance from 2002 to 2016

**DOI:** 10.3390/microorganisms9050963

**Published:** 2021-04-29

**Authors:** Jiun-Ling Wang, Wen-Chien Ko, Chih-Hsin Hung, Ming-Fang Cheng, Hui-Ying Wang, Yih-Ru Shiau, Jui-Fen Lai, I-Wen Huang, Li-Yun Hsieh, Tsai-Ling Yang Lauderdale

**Affiliations:** 1Department of Internal Medicine, National Cheng Kung University Hospital, Tainan 704, Taiwan; jiunlingwang@gmail.com (J.-L.W.); winston3415@gmail.com (W.-C.K.); 2Department of Medicine, National Cheng Kung University, Tainan 704, Taiwan; 3Department of Chemical Engineering and Institute of Biotechnology and Chemical Engineering, Shou University, Kaohsiung 840, Taiwan; chhung@cloud.isu.edu.tw (C.-H.H.); mfcheng@vghks.gov.tw (M.-F.C.); 4Department of Pediatrics, Kaohsiung Veterans General Hospital and School of Medicine, Kaohsiung 813, Taiwan; 5School of Medicine, National Yang-Ming University, Taipei 112, Taiwan; 6Department of Nursing, Fooyin University, Kaohsiung 831, Taiwan; 7National Institute of Infectious Diseases and Vaccinology, National Health Research Institutes, Zhunan 350, Taiwan; iris@nhri.edu.tw (H.-Y.W.); yihru@nhri.edu.tw (Y.-R.S.); juifen@nhri.edu.tw (J.-F.L.); yvone@nhri.edu.tw (I.-W.H.); lilianfly@nhri.edu.tw (L.-Y.H.)

**Keywords:** ST131 clone, *E. coli*, community onset, urinary tract infection, population-base surveillance

## Abstract

Sequence type (ST) 131 is a multidrug-resistant pandemic lineage of *E. coli* responsible for extraintestinal infections. Few surveillance data of ST131 included all antimicrobial-susceptible and -resistant isolates or focused on community-acquired urinary tract infection (UTI). From a population-based surveillance pool of 2997 outpatient urine *E. coli* isolates, 542 were selected for detection of ST131 based on ciprofloxacin and/or cefotaxime resistance. Pulsed-field gel electrophoresis (PFGE) was performed on all ST131 isolates to further determine their relatedness. The estimated overall ST131 prevalence in this community UTI cohort increased from 11.2% (in 2002–2004), 12.2% (in 2006–2008), 13.6% (in 2010–2012), to 17.4% in 2014–2016 (*p* < 0.01). In the ciprofloxacin-resistant/cefotaxime-resistant group, ST131 increased from 33.3% in 2002–2004 to 72.1% in 2014–2016 (*p* < 0.01). In the ciprofloxacin-resistant/cefotaxime-susceptible group, ST131 was found in 24.3% overall without significant increase in its prevalence over time. PFGE showed emergence of a cluster of ciprofloxacin-resistant/cefotaxime-resistant ST131 carrying Gr. 1 CTX-M ESBL in 2014–2016, especially 2016. Multivariate analysis revealed that age (≥65 y.o) and ciprofloxacin resistance were independent factors associated with ST131. This longitudinal surveillance showed that ciprofloxacin-resistant/cefotaxime-susceptible ST131 has been circulating in the community since 2002 but ciprofloxacin-resistant/cefotaxime-resistant ST131 increased rapidly in the later years.

## 1. Introduction

*Escherichia coli* is the most common cause of community-acquired urinary tract infections (UTI) [1]. In recent years, there has been an increase of extended-spectrum β-lactamase (ESBL)-producing and fluoroquinolone-resistant *E. coli* in community infections in many areas including Europe, America, and Asia. For example, the incidence of antibiotic-resistant *E coli* bacteriuria nearly doubled from 2005 to 2009 among elderly patients and those with community-associated isolates in a population-based study in the USA [2]. In a 12-year longitudinal study in Spain (1999–2010), the proportion of *E. coli* isolates resistant to third-generation cephalosporins and fluoroquinolones increased significantly (from 5% to 15% and from 16% to 30%, respectively) [3]. In an earlier report from our Taiwan Surveillance of Antimicrobial Resistance (TSAR) program on *E. coli* from outpatients, the proportion of isolates non-susceptible to fluoroquinolone and the prevalence of ESBL-producers increased from 15.0% and 4% in 2002 to 26.9% and 11.7% in 2012, respectively, with the ESBL being nearly all (96.3%) due to CTX-M type [4]. The emergence of *E. coli* strains resistant to extended-spectrum cephalosporins and fluoroquinolones is a public health threat in both developed and developing countries [5].

Based on multilocus sequence typing (MLST), the predominant lineages of *E. coli* causing extraintestinal infections are sequence type (ST) 69, 73, 95, 127, and 131, among which ST131 is the most multidrug-resistant [6]. ST131 *E. coli* was first reported in 2008 but later shown to have emerged in late 1990s [7,8,9]. Since then, ST131 *E. coli* spread rapidly to become a dominant pandemic clone worldwide, in part because most ST131 strains are ESBL-producers and/or fluoroquinolone-resistant [10]. UTIs caused by CTX-M ESBL-positive ST131 are now widespread in the community, hospital, and long-term care facility [11,12]. In a systemic review on extraintestinal pathogenic *E. coli* linages, ST131 occupied the highest summary proportions globally, especially in studies that selected for resistant isolates, with an estimated 30% prevalence among ESBL-producing and fluoroquinolone-resistant *E. coli* isolates [13].

Although there have been a few longitudinal population-based studies on the role of ST131 clone in ESBL-producing *E. coli* or fluoroquinolone-resistant *E. coli* in bacteremia [14,15,16], there is little longitudinal multicenter surveillance data on ST131 in community UTI. Taking advantage of isolates collected systematically in a surveillance program over a 14-year span [4,17], the present study was carried out to better understand the role of ST131 in community UTI in Taiwan. We first grouped the *E. coli* isolates from outpatient urine samples by their combined fluoroquinolone and extended-spectrum cephalosporin susceptibility to investigate the prevalence and trend of the four predominant extraintestinal *E. coli* lineages in different time period, then focused on ST131 molecular epidemiology.

## 2. Materials and Methods

### 2.1. Isolates

*E. coli* isolates used in this study are from the Taiwan Surveillance of Antimicrobial Resistance (TSAR) program between 2002 and 2016 [4]. The isolates were part of the TSAR biennial collection from a total number of 28 hospitals located in different geographic regions of Taiwan. Detailed isolate collection protocol and TSAR participating hospitals have been described elsewhere [4,18]. The TSAR project was approved by the Research Ethics Committee of NHRI (EC960205-E, EC1010602-E, EC1030406-E, and EC1050606-E). Written informed consent was not obtained because the study only used bacterial isolates recovered from clinical samples taken as part of standard care and patient information was anonymized and de-identified prior to analysis. All isolates were stored at −70°C for subsequent testing. The present study focused on isolates recovered from urine samples of outpatient and emergency rooms (hereafter referred to as outpatient).

### 2.2. Antibiotic Susceptibility Testing (AST) and Data Analysis

For AST, broth microdilution method was performed following the guidelines of CLSI and the instructions of the manufacturer to determine minimum inhibitory concentrations (MIC) of different antimicrobial agents using custom designed or standard Sensititre panels (ThermoFisher Scientific (formerly Trek Diagnostics), East Grinstead, UK). Interpretive criteria are based on the current CLSI breakpoints except for ciprofloxacin, for which the old (2018) and current (since 2019) breakpoints were applied [19,20]. Susceptibility to cefazolin was determined using breakpoints for uncomplicated urinary tract infections (UTI).

Susceptibility interpretation analysis was made using the WHONET software [21]. Duplicate isolates were excluded from analysis. Due to the reported high association of ST131 with fluoroquinolone and/or extended-spectrum β-lactam resistance, we first categorized the isolates into 4 groups based on their combined fluoroquinolone (represented by ciprofloxacin (CIP)) and extended-spectrum cephalosporin (represented by cefotaxime (CTX), the preferred substrate of CTX-M ESBL) susceptibilities. The four groups of isolates included CIP-R/CTX-R, CIP-R/CTX-S, CIP-S/CTX-R, and CIP-S/CTX-S (S for susceptible, R for resistant plus intermediate). Cefotaxime was chosen to represent extended-spectrum cephalosporin because studies have found CTX-M to be the most common ESBL in *E. coli* from Taiwan and in UTI causing ST131 *E. coli* [4,11,12]. For CIP, the 2018 CLSI breakpoints were used in this grouping.

### 2.3. ESBL and AmpC β-Lactamase Gene Detection

Isolates with aztreonam, ceftazidime, or cefotaxime MIC ≥ 2 mg/L were subject to PCR for the presence of genes encoding CTX-M group 1, 2, 8, and 9 and SHV-type ESBL plus CMY- and DHA-type AmpC β-lactamases. Multiplex PCR assays were carried out following previously described primers and protocols [22,23].

### 2.4. Multiplex PCR for Sequence Type (ST) 131, 69, 73, and 95 Determination

To determine the prevalence and trend of ST131 *E. coli* in outpatient UTI, a multiplex PCR assay to identify isolates belonging to the above mentioned four STs was used [6]. Due to the large number of outpatient urine isolates in the collection, we selected from each hospital at least one isolate fitting the four CIP/CTX groups described above (if available) in each study year. As there were very few CIP or CTX intermediate isolates, they were not included in this selection. For ease of comparison, isolates were grouped into 4 periods: 2002–2004, 2006–2008, 2010–2012, and 2014–2016. A total of 542 isolates were selected including 116, 126, 150, and 150 isolates from 2002–2004, 2006–2008, 2010–2012, and 2014–2016, respectively.

### 2.5. Pulsed-Field Gel Electrophoresis Analysis (PFGE)

To further determine the clonal relatedness of ST131 isolates from different years and hospitals, PFGE was performed on all ST131 isolates using XbaI-digested DNA following a previously described protocol [24]. A PFGE dendrogram was generated using the BioNumerics software based on Dice coefficients. A Dice similarity index ≥80% was considered to belong to the same PFGE cluster.

### 2.6. Statistical Analysis

Chi square for trend was used for changes in distribution of the four CIP/CTX susceptibility profile groups using EpiInfo from the USA CDC (https://www.cdc.gov/epiinfo/). To identify factors associated with ST131, categorical variables were compared first using Chi-square or Fisher’s exact test. Multivariable logistic regression analysis was then performed using the study period, hospital geographic region, patient age group, ciprofloxacin-susceptibility, and cefotaxime-susceptibility as variables using SAS 9.2 (SAS Institute, Cary, NC, USA). A 2-tailed *p value* of <0.05 was considered statistically significant.

## 3. Results

A total of 2997 outpatient urine *E. coli* isolates were in the TSAR biennial collection from 2002 to 2016 with over 600 isolates in each study period. Among the 2997 isolates, 380 (12.7%) were CIP-R/CTX-R, 383 (12.8%) were CIP-R/CTX-S, 130 (4.3%) were CIP-S/CTX-R, and 2104 (70.2%) were CIP-S/CTX-S (based on 2018 CLSI breakpoints. Isolates with intermediate results are grouped in the resistance category due to the fewer number of those isolates). However, a steady increase of isolates that were CIP-R/CTX-R occurred over time, from 4.8% in 2002–2004 to 17.8% in 2004–2016 (*p* < 0.01) with a concurrent decrease of CIP-S/CTX-S isolates (Figure 1). CIP-R/CTX-S isolates comprised similar percentages (~11%) before 2012 but increased to 16.6% in 2014–2016. There were no significant changes in the proportion of CIP-S/CTX-R isolates over the years. Not surprisingly, isolates of the CIP-R/CTX-R and CIP-S/CTX-R groups had the lowest rates of susceptibility to other β-lactam agents. Among these two CTX-R groups, the CIP-R isolates had the lowest rates of susceptibility to aztreonam, cefepime, gentamicin, and trimethoprim/sulfamethoxazole. In the CTX-S groups, the CIP-R isolates also had lower rates of susceptibility to ampicillin, cefoxitin, gentamicin, and trimethoprim/sulfamethoxazole (Table 1).

The 542 isolates that were selected to investigate for distribution of the 4 major *E. coli* lineages (ST131, ST69, ST73, and ST95) associated with extraintestinal infections included 167 CIP-R/CTX-R, 148 CIP-R/CTX-S, 36 CIP-S/CTX-R, and 191 CIP-S/CTX-S isolates. In the CIP-R/CTX-R group, ST131 predominated (45.5% overall, 76/167) and its prevalence increased from 33.3% (7/21) in 2002–2004 to 72.1% (31/43) in 2014–2016 (*p* < 0.01) (Figure 2). In the CIP-R/CTX-S group, ST131 was detected in 24.3% (36/148) overall but there was no significant increase in its prevalence in this group of isolates over time. ST131 was detected in a small portion of the CIP-S/CTX-R (5.6%, 2/36) and CIP-S/CTX-S (5.8%, 11/191) groups.

In contrast, the three non-ST131 major ST lineages investigated were found mostly in the CIP-S/CTX-S group. ST69 was detected in 5.2% (10/191) of the CIP-S/CTX-S group and only in a small proportion (0.6–2.8%) of the other 3 CIP/CTX groups. ST73 was detected only in the CIP-S/CTX-S group (11.0%, 21/191). ST95 was detected in 27.2% (52/191) of the CIP-S/CTX-S group and a small portion of the CIP-R/CTX-R (3.0%, 5/167) and CIP-S/CTX-R (2.8%, 1/36) groups. Together, these results indicated that ST131 is present in the majority of our CIP-R/CTX-R *E. coli* isolates while ST69, ST73, and ST95 are mostly found in the CIP-S/CTX-S isolates.

We then compared the antibiograms of these 4 STs (Table 2). As expected, ST131 isolates had the lowest rates of susceptibility to all β-lactam agents tested (except carbapenems) and fluoroquinolones compared to the other three STs. ST131 isolates were significantly less susceptible to ciprofloxacin (10.4 vs. 78.6–100%, *p* < 0.01). They also had ~2-fold lower susceptibility to cephalosporins cefazolin (33.6 vs. 85.7–100%), cefuroxime (34.4 vs. 78.6–100%), cefotaxime (37.6 vs. 85.7–100%), and cefepime (48.8 vs. 85.7–100%). In addition, less than half of the ST131 isolates were susceptible to gentamicin (48.0%) and trimethoprim/sulfamethoxazole (40.8%).

Among the 542 isolates, 205 had aztreonam, ceftazidime, and/or cefotaxime MIC > = 2 μg/mL, including 79 ST131 isolates and 126 non-ST131 isolates (2 ST69, 6 ST95, and 118 ST-unknown). These 205 isolates were subject to ESBL and AmpC gene determination. CTX-M-type ESBL gene was detected in 83.5% (66/79) of the ST131 isolates and 66.7% (84/126) of the non-ST131 isolates. SHV-type ESBL gene was detected in 6 isolates including one ST131 and 5 ST-unknown isolates. All CTX-M belonged to either Gr. 1 or Gr 9. However, Gr. 9 CTX-M predominated in 2002–2012 (75/100, 75%), but all 13 CTX-M-positive ST131 from 2014–2016 carried Gr. 1 (including 1 isolate that was positive for both Gr. 1 and Gr. 9). AmpC β–lactamase was detected in 31.6% (25/79) of the ST131 and 41.3% (52/126) of the non-ST131 isolates tested, 16.5% (13/79) and 12.7% (16/126) of which, respectively, were also ESBL-positive.

Using data on the number of total and CIP-R/CTX-R, CIP-R/CTX-S, CIP-S/CTX-R, and CIP-S/CTX-S group isolates in each study period in Figure 1, plus the percentages of ST131 obtained in the tested isolates from each CIP/CTX group in Figure 2, we calculated the prevalence of ST131 in outpatient UTI *E. coli* in each study period. For example, among the 844 isolates in 2014–2016, 151 (17.9%), 140 (16.5%), 41 (4.9%), and 512 (60.7%) had CIP-R/CTX-R, CIP-R/CTX-S, CIP-S/CTX-R, and CIP-S/CTX-S phenotype, respectively; and the percentage of ST131 in the tested isolates of each group was 72.1%, 20.0%, 0%, and 2.0%, respectively. The estimated ST131 prevalence in *E. coli* in this cohort of isolates is 13.9% overall (2002 to 2016) and 17.4% in 2014–2016, which was a significant increase (*p* < 0.01) from the estimated prevalence of 11.2% in 2002–2004, 12.2% in 2006–2008, and 13.6% in 2010–2012.

PFGE was performed on the ST131 isolates to look for strain relatedness in isolates from different years and hospitals (Figure 3). Among the 125 isolates, four main clusters (isolates with ≥80% similarity) predominated, including 33 isolates in cluster A, 15 isolates in cluster B, 11 isolates in cluster C, and 10 isolates in cluster D (Figure 3). These isolates are from different periods, geographic regions, and hospitals. The 4 clusters also shared ≥75% in similarity. All but one of the 33 isolates in cluster A are from 2002–2012, and the majority (22 isolates) of them were CIP-R/CTX-S. Cluster A isolates also contain 8 CIP-R/CTX-R isolates, 7 of which carried Gr. 9 CTX-M type ESBL. In contrast, in the 36 isolates belonging to clusters B, C, and D, all but 2 were CIP-R/CTX-R, and most are from 2014–2016, but Gr. 9 CTX-M predominated in cluster B, Gr. 1 CTX-M in cluster C, and all cluster D isolates carried Gr. 9 CTX-M. Isolates outside of these 4 clusters are mostly diverse in their CIP/CTX susceptibility phenotype.

Univariate and multivariate analysis was performed on the 542 isolates to identify factors associated with ST131 (Table 3). In univariate analysis, there was no difference in the geographic region between ST131 and non-ST131 isolates, but the ST131 isolates were more commonly found in > = 65 y old patients, in ciprofloxacin-resistant isolates, and in cefotaxime-resistant isolates. Multivariate analysis revealed that > = 65 y old patients (OR 1.739; 95% CI 0.42–3.089), and ciprofloxacin resistance (OR 26.769; 95% CI 3.458–207.191), but not cefotaxime resistance, remained independent factors associated with ST131.

## 4. Discussion

ST131 is a dominant multidrug-resistant lineage of extraintestinal pathogenic *E. coli* responsible for urinary tract infections (UTI) and bacteremia [10,12]. Although there have been many reports on ST131 from different countries, they mostly focused on fluoroquinolone-resistant and/or ESBL-producers, particularly of the CTX-M-type, and did not separate inpatient and outpatient isolates [13]. In Taiwan, the few studies on ST131 *E. coli* to date focused on bacteremia isolates and mostly on ESBL-producers except for one report [25,26,27,28,29]. Utilizing a biennial nationwide surveillance cohort of outpatient UTI isolates from 2002 and 2016, the present study investigated the role of and changes in ST131 *E. coli* causing community UTI in Taiwan.

When we grouped our isolates based on their co-resistance to fluoroquinolones (FQ), ciprofloxacin (CIP), and extended-spectrum cephalosporin (ESC) cefotaxime (CTX), we found a significant increase of CIP-R/CTX-R isolates, from 4.8% in 2002–2004, to 15.1% in 2010–2012, and 17.9% in 2014–2016 (*p* < 0.01). A similar trend was found in *E. coli* from hospitals in Spain, where the proportion of isolates with co-resistance to FQ and ESC increased from 1.6% in 1999 to 11.3% in 2010 [3]. In the present study, ST131 was most prevalent in the CIP-R/CTX-R isolates (45% overall) with a concurrent significant increase in its proportion in later years (29.5–33.3% in 2002–2008 vs. 42.4%–72.1% in 2010–2016). In contrast, although CIP-R/CTX-S isolates had the second highest overall proportion of ST131 (24.3%), no steady increase in its prevalence was observed over the years, while ST131 comprised only a small proportion (5.6–5.8%) of the CIP-S/CTX-R and CIP-S/CTX-S isolates. The trend of increasing CIP-R ST131 and CTX-R ST131 in the present study is similar to a study on WHO’s archival collection of *E. coli* (1957 to 2011) showing progressively more antimicrobial resistance and ST131 isolates [30].

The estimated overall prevalence of ST131 in our cohort of community UTI *E. coli* isolates from 2002 to 2016 was 13.9%, with an increase from 11.2% in 2002–2004 to 17.4% in 2014–2016 (*p* < 0.01). The increase was attributed to the increase of isolates that are co-resistant to FQ and ESC plus the higher proportion of ST131 among these isolates in later years. In a review of studies on extraintestinal *E. coli* lineages between 1998 and 2018 from all the different regions, the estimated overall summary proportions of ST131 was 24% (95% CI: 18–30%) for urine isolates without differentiating those from inpatients and outpatients [13]. The prevalence of ST131 in Taiwan could be higher since ST131 has been reported in higher proportion of healthcare-associated isolates than community-associated isolates in other countries [31]. The increase of ST131 in our community UTI isolates is a concern since 90%, 62.4%, 52.0%, and 59.2% of them are resistant to ciprofloxacin, cefotaxime, gentamicin, and trimethoprim/sulfamethoxazole, respectively.

Among the 4 main PFGE clusters of ST131 isolates, the majority of isolates in cluster A were from earlier years, and most were CIP-R/CTX-S, with the few CIP-R/CTX-R isolates carrying Gr. 9 CTX-M-type ESBL. In contrast, ST131 isolates in clusters B, C, and D were mostly from later years and nearly all were CIP-R/CTX-R, but Gr. 9 CTX-M-carrying isolates predominated in clusters B and D, while Gr. 1 CTX-M-carrying isolates predominated in cluster C. Most Gr. 1 CTX-M isolates were from 2014–2016, especially 2016. These results indicated a decrease of Gr. 9 CTX-M and increase of Gr. 1 CTX-M ST131 isolates in Taiwan. An updated study on bacteremia isolates from Canada also found an increase of CTX-M-15 (a Gr. 1 CTX-M) clade C2 isolates with a concurrent decrease of CTX-M-14 (a Gr. 9 CTX-M) [32]. Of note, isolates of the four main PFGE clusters comprised more than half (55.2%) of the ST131 in our study. They also shared 75% in similarity and thus may be less genetically diverse. These isolates were from hospitals located in different regions of Taiwan indicating possible spread and evolution of a particular subclone. Further studies are needed to determine if these isolates belong to a predominate subclone/clade of ST131 [10,32,33].

Our cohort showed that being elderly is an independent risk factor of ST131 infection. Host-specific factors or healthcare-associated transmission may be the reason why ST131 were more likely found in the elderly group. From studies conducted in the UK and Ireland, ST131 was commonly found in long-term care facilities [34,35]. A study from the US also found the prevalence of *E. coli* ST131 infection increased with age [31]. In a Hong Kong cohort, the percentage of ST131 was higher among isolates from ≥65 years old patients than those from under 65 years old [36]. Another fecal colonization study in the US also showed long-term care facilities to be reservoirs for antimicrobial-resistant ST131 *E. coli* [37]. Thus, the finding that age is an independent factor correlated with ST131 UTI is not unexpected.

Although univariate analysis showed that fluoroquinolone resistance and extended-spectrum cephalosporin (cefotaxime) resistance were factors associated with ST131, only fluoroquinolone resistance remained an independent factor in multivariate analysis. In Taiwan, the emergence and increase of fluoroquinolone resistance has been found in different community pathogens, including *Haemophilus influenzae,* and Gr. A and Gr. B *Streptococcus* [18,38], which likely resulted from under documented fluoroquinolone use in the community setting [39]. Finding of FQ resistance being an independent factor associated with ST131 further highlights the need for close monitoring of community fluoroquinolone use to prevent the spread of multidrug-resistant ST131 strains in Taiwan.

## 5. Conclusions

This longitudinal national surveillance showed that the prevalence of ST131 in a cohort of community UTI isolates increased steadily over the years. We also found that ciprofloxacin-resistant/cefotaxime-susceptible ST131 has been circulating in the community since 2002 without a steady increase in its prevalence. In contrast, ciprofloxacin-resistant/cefotaxime-resistant ST131 increased rapidly in the later years starting in 2010. A cluster of ST131 that was co-resistant to ciprofloxacin and cefotaxime and carried Gr. 1 CTX-M emerged in 2014–2016, especially 2016. Being elderly and fluoroquinolone resistance were independent risk factors of ST131 infection in this population-based surveillance. These results indicate that continued monitoring of ST131 prevalence to detect emerging subclones with different antimicrobial resistance profiles and its association with community antibiotic consumption is warranted.

## Figures and Tables

**Figure 1 microorganisms-09-00963-f001:**
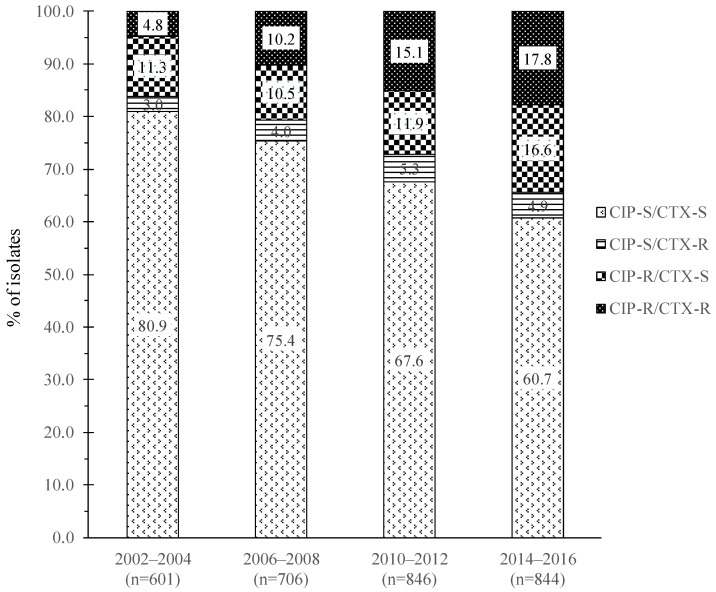
Proportion of outpatient urine *E. coli* with co-resistance to fluoroquinolone and extended-spectrum cephalosporin. CIP, ciprofloxacin; CTX, cefotaxime; S, susceptible; R, resistant.

**Figure 2 microorganisms-09-00963-f002:**
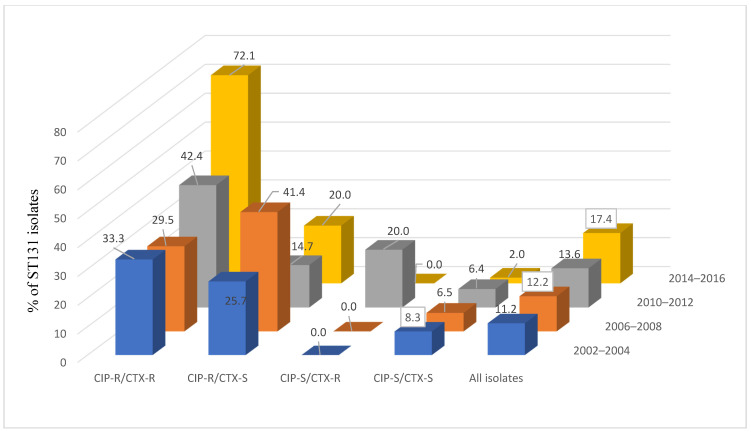
Prevalence of ST131 in *E. coli* isolates with different fluoroquinolone and extended-spectrum cephalosporin co-resistance profiles in different study years. CIP, ciprofloxacin; CTX, cefotaxime; S, susceptible; R, resistant.

**Figure 3 microorganisms-09-00963-f003:**
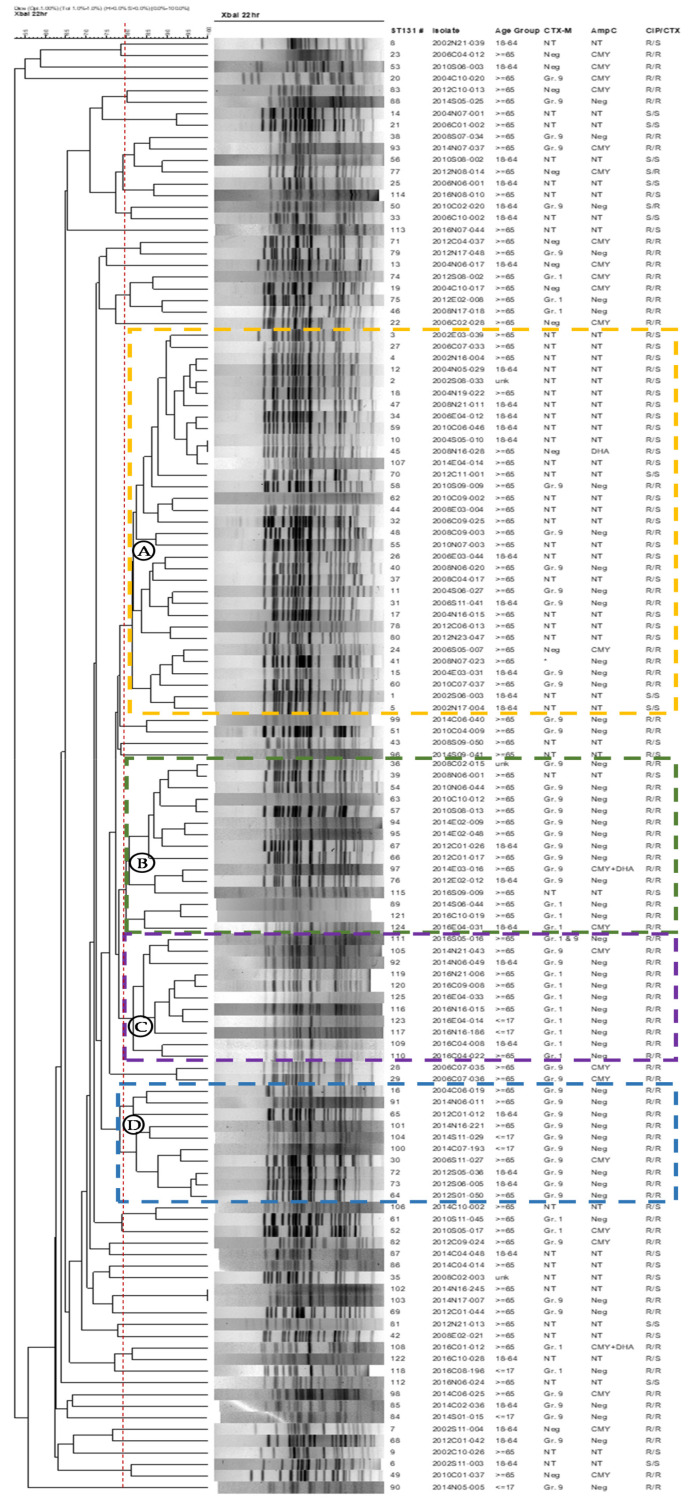
Dendrogram of 125 ST131 *E. coli* isolates based on pulsed-field gel electrophoresis analysis. Isolate, the first 4 digits indicate isolation year, followed by region [C (central), N (northern), E (eastern), S (southern)], 2 digits of hospital number, and 3 digits of isolate number. Gr., group of CTX-M ESBL detected; * SHV-type ESBL detected. NT, not tested. CIP, ciprofloxacin; CTX, cefotaxime; S, susceptible; R, resistant.

**Table 1 microorganisms-09-00963-t001:** Susceptibility to different agents in 2997 outpatient urine *E. coli* isolates grouped by their fluoroquinolone and extended-spectrum cephalosporin co-resistance profile ^a^, 2002–2016 combined.

Antimicrobial Agent ^b^	CIP-R/CTX-R (N = 380)			CIP-R/CTX-S (N = 383)			CIP-S/CTX-R (N = 130)			CIP-S/CTX-S (N = 2104)		
	%S	%I	%R	%S	%I	%R	%S	%I	%R	%S	%I	%R
Amikacin	93.9	1.6	4.5	99.7	0.3	0	96.1	0.8	3.1	100	0	0
Ampicillin	0	0	100	14.6	0.5	84.9	0	0	100	39.9	0.2	59.9
Aztreonam	16.1	22.6	61.3	99.7	0	0.3	29.2	33.1	37.7	100	0	0
Cefazolin–UTI	0.3	-	99.7	92.3	-	7.7	2.3	-	97.7	96.4	-	3.6
Cefepime	39.7	15.3	45.0	100	0	0	64.6	10.8	24.6	100	0	0
Cefotaxime	0	0.8	99.2	100	0	0	0	3.8	96.2	100	0	0
Cefoxitin	30.3	10.8	58.9	78.6	16.4	5.0	30.0	3.8	66.2	97.4	1.9	0.7
Ceftazidime	27.6	9.5	62.9	100	0	0	24.6	16.9	58.5	99.9	0.1	0
Cefuroxime	0.5	2.1	97.4	85.1	12.3	2.6	1.5	3.9	94.6	97.3	2.3	0.4
Ciprofloxacin	0	0	100	0	0	100	80.8	11.5	7.7	92.5	4.8	2.7
Ciprofloxacin (2018)	0	1.6	98.4	0	2.3	97.7	100	0	0	100	0	0
Gentamicin	38.9	1.1	60.0	53.3	0.8	46.0	62.3	3.1	34.6	86.4	2.0	11.6
Imipenem	98.4	1.1	0.5	100	0	0	99.2	0.8	0	100	0	0
Trimethoprim/sulfa.	30.0	-	70.0	36.6	0	63.4	40.0	-	60.0	58.3	-	41.7

^a^ In the initial CIP (ciprofloxacin)/CTX (cefotaxime) grouping, intermediate results were included in the resistant category, and the old (2018) CLSI breakpoints for CIP were used. ^b^ Percentages shown are based on current CLSI breakpoints, except ciprofloxacin, for which the old (2018) and current breakpoints were used. For cefepime, the SDD (susceptible dose dependent) results are reported as I (intermediate). For cefazolin, breakpoints for uncomplicated urinary tract infections (UTI) were applied; S, susceptible; R, resistant.

**Table 2 microorganisms-09-00963-t002:** Antimicrobial susceptibility of ST131 vs. ST69, ST73, and ST95 *E. coli* isolates.

Antimicrobial Agent ^a^	ST131 (N = 125)			ST69 (N = 14)			ST73 (N = 21)			ST95 (N = 58)		
	%S	%I	%R	%S	%I	%R	%S	%I	%R	%S	%I	%R
Amikacin	96.0	2.4	1.6	100	0	0	100	0	0	98.3	0	1.7
Ampicillin	3.2	0	96.8	14.3	0	85.7	52.4	0	47.6	36.2	0	63.8
Aztreonam	50.4	16.0	33.6	85.7	14.3	0	100	0	0	93.1	3.4	3.4
Cefazolin–UTI	33.6	-	66.4	85.7	0	14.3	100	-	0	87.9	-	12.1
Cefepime	48.8	14.4	36.8	85.7	0	14.3	100	0	0	94.8	0	5.2
Cefotaxime	37.6	0	62.4	85.7	0	14.3	100	0	0	89.7	0	10.3
Cefoxitin	65.6	10.4	24.0	92.9	7.1	0.0	100	0	0	93.1	1.7	5.2
Ceftazidime	64.0	5.6	30.4	100	0	0	100	0	0	93.1	1.7	5.2
Cefuroxime	34.4	3.2	62.4	78.6	7.1	14.3	100	0	0	89.7	0	10.3
Ciprofloxacin	9.6	0	90.4	64.3	14.3	21.4	100	0	0	91.4	0	8.6
Ciprofloxacin (2018)	10.4	0	89.6	78.6	0	21.4	100	0	0	91.4	0	8.6
Gentamicin	48.0	0	52.0	64.3	0	35.7	90.5	-	9.5	87.9	0	12.1
Trimethoprim/Sulfa.	40.8	-	59.2	28.6	-	71.4	85.7	-	14.3	53.4	-	46.6

^a^ Based on current CLSI breakpoints except ciprofloxacin, for which both the old (2018) and current breakpoints were used. Breakpoints for uncomplicated urinary tract infections (UTI) were used for cefazolin. For cefepime, the SDD (susceptible dose dependent) results are reported as I (intermediate). All isolates were susceptible to ertapenem and imipenem.

**Table 3 microorganisms-09-00963-t003:** Factors associated with ST131 in *E. coli* from outpatient UTI.

Strata	ST131	Non-ST131	*p* ^a^	OR ^b^	95%CI ^b^	*p* ^b^
Number of isolates	125	417				
Geographic region			NS			
Central	47 (37.6)	129 (30.9)				
Eastern	15 (12.0)	45 (10.8)				
Northern	35 (28.0)	115 (27.6)				
Southern	28 (22.4)	128 (30.7)				
Age ^c^			0.001			
18–64	31 (24.8)	185 (44.4)		Reference		
> = 65	84 (51.3)	194 (46.5)		1.739	1.058–2.858	0.029
< = 17	7 (5.6)	29 (7.0)		1.139	0.368–7.652	NS
Ciprofloxacin						
Susceptible	13 (10.4)	214 (51.3)	<0.001			
Resistant	112 (89.6)	203 (48.7)		26.769	3.458–207.191	0.002
Cefotaxime			<0.001			
Susceptible	47 (37.6)	292 (70.0)				
Resistant	78 (62.4)	125 (30.0)		1.217	0.556–2.664	0.622

^a^ *p value* by chi-square test, NS, not significant, ^b^ *p value* by multivariate analysis; OR, odds ratio; CI, confidence interval, ^c^ Age of source patient was unknown in 12 isolates (3 in ST131, 9 in non-ST131 group).

## Data Availability

Please contact author for data request.

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
