# Peer review of "Temporal Trend of ST131 Clone among Urinary Escherichia coli Isolates in the Community: A Taiwan National Surveillance from 2002 to 2016"

_microorganisms, 2021, doi:10.3390/microorganisms9050963_

Round 1
Reviewer 1 Report
It is a very large and very interesting study, but it also has some important limitations. Thus, the clades and subclades of the ST131 strains, nor the type of ESBL enzyme, have not been established. Furthermore, the 542 selected isolates have not been chosen proportionally. However, the study has enough merit to justify its publication.
AmpC β–lactamase was detected in a total of 77 (61.6%) ST131 isolates? (line 200).
Reviewer 2 Report
The authors should discuss why they chose outpatient samples, and how the isolates there differ from admitted patient samples.
Fig 3 should be submitted at a better resolution as the current picture is blurry at higher magnification and not at all readable.
Also, there are some things such as line 295 "being elderly" should be used instead of "elderly". Line 302 "the finding that age is an independent factor correlated with ST131" should be sued instead of the phrase used presently.
